# Alterations in Immune Response Profile of Tumor-Draining Lymph Nodes after High-Intensity Focused Ultrasound Ablation of Breast Cancer Patients

**DOI:** 10.3390/cells10123346

**Published:** 2021-11-29

**Authors:** Xue-Qiang Zhu, Pei Lu, Zhong-Lin Xu, Qiang Zhou, Jun Zhang, Zhi-Biao Wang, Feng Wu

**Affiliations:** 1Institute of Ultrasonic Engineering in Medicine, Chongqing Medical University, Chongqing 400016, China; zhuxueqiang1234@163.com (X.-Q.Z.); hnlylpnh@gmail.com (P.L.); xuzhonglin0405@163.com (Z.-L.X.); zhouqiang176@163.com (Q.Z.); zhj316316@163.com (J.Z.); wangzhibiao@haifu.com.cn (Z.-B.W.); 2Cancer Center, Sichuan Academy of Medical Sciences & Sichuan Provincial People’s Hospital, Chengdu 610072, China; 3Department of Oncology, Nanyang First People’s Hospital, Nanyang 473004, China; 4Nuffield Department of Surgical Sciences, University of Oxford, Oxford OX3 9DU, UK

**Keywords:** high-intensity focused ultrasound, breast cancer, immunomodulation, ablation, lymph node, immune response, cytotoxic T lymphocyte, natural killer cell, immunotherapy

## Abstract

Previous studies have revealed that high-intensity focused ultrasound (HIFU) ablation can trigger an antitumor immune response. The aim of this study was to investigate immune response in tumor-draining lymph nodes (TDLNs) after HIFU treatment. Forty-eight female patients with biopsy-confirmed breast cancer were divided into a control group and an HIFU group. In the control group, 25 patients underwent modified radical mastectomy, but 23 patients in the HIFU group received HIFU ablation of primary cancer, followed by the same operation. Using HE and immunohistochemical staining, the immunologic reactivity pattern and immune cell profile were assessed in paraffin-embedded axillary lymph nodes (ALNs) in all patients. The results showed that ALNs presented more evident immune reactions in the HIFU group than in the control group (100% vs. 64%). Among the ALNs, 78.3% had mixed cellular and humoral immune response, whereas 36% in the control group showed cellular immune response. The numbers of CD3^+^, CD4^+^, NK cell, and activated CTLs with Fas ligand^+^, granzyme^+^ and perforin^+^ expression were significantly higher in the ALNs in the HIFU group. It was concluded that HIFU could stimulate potent immune response and significantly increase T cell, activated CTLs and NK cell populations in the TDLNs of breast cancer.

## 1. Introduction

The lymphatic system constitutes an essential compartment of the immune system. It transports antigens and immune cells from peripheral tissues to draining lymph nodes, and from there back into blood circulation. Tumor-draining lymph nodes (TDLNs) are peripheral secondary lymphoid organs that play an important role in host antitumor immunity. As the first line of host defense against the dissemination of tumor cells, they serve as initial and effective barriers to prevent cancer cells spreading from the primary tumor to distant organs [1]. TDLNs are composed of different types of immune cells, including antigen presenting cells (APCs), natural killer (NK) cells and T and B lymphocytes. They are the primary site for tumor antigen presentation and lymphocyte activation through their organized compartments, and initiate a TDLN-mediated immune response in the antitumor immunity [2]. T cell subpopulations that are activated during antitumor immune response decide the outcome of the interaction between host and tumor. Immune response in the TDLNs may be a B-cell predominance that is characterized morphologically by follicular hyperplasia, or a T-cell predominance with a characteristic pattern of T-cell hyperplasia [3,4,5,6].

As a noninvasive approach, high-intensity focused ultrasound (HIFU) treatment has been used in the clinical management of tumor patients. It is usually performed for patients with a solid tumor, intending to destroy all sites of the disease. In addition to local ablation, numerous studies have revealed that HIFU can trigger host immune responses [7,8,9,10,11,12,13]. It is becoming increasingly evident that HIFU local therapy can have systemic antitumor immune effects, which may help the immune system reduce local recurrence and control metastases. In our previous studies related to HIFU treatment for patients with breast cancer, we found clinical evidence that HIFU could induce immunological effects on APCs and tumor-infiltrating lymphocytes (TILs) in the primary cancer [14,15]. The tumors ablated with HIFU prior to a surgical intervention presented APCs and TILs localized along the periphery margin of the ablation zone. Compared to the tumors that underwent a surgical resection alone, we observed a significant increase in dendritic cells, macrophages and B lymphocytes, as well as CD3, CD4, CD8, CD4/CD8, NK cells and activated cytotoxic T lymphocytes (CTLs) in HIFU-treated breast cancer. However, to our knowledge, little is known about the consequences of HIFU ablation on TDLNs’ immune status and function. There is no evidence to indicate whether HIFU ablation for primary cancer can modify antitumor immune response in the TDLNs, or if these modifications in the TDLNs might have any clinical significance such as an influence on patient outcome. The aim of the current study, therefore, was to explore the immune status and immune cell profile of the TDLNs after HIFU treatment for breast cancer, and to provide better understanding of the HIFU-enhanced antitumor immunity.

## 2. Materials and Methods

### 2.1. Patients

The original trial was carried out from April 1998 to August 1999. It was designed to explore the safety, efficacy and feasibility of HIFU ablation in patients with localized breast cancer. Forty-eight women with biopsy-confirmed breast cancer were enrolled into a prospective clinical trial, and the details of the trial have been published [14]. Briefly, the patients were randomly divided into a control group (*n* = 25) and an HIFU group (*n* = 23). They all underwent modified radical mastectomy. But in the HIFU group, HIFU treatment for breast lesion was performed 1–2 weeks prior to the mastectomy. The trial was approved by the ethics committee of Chongqing Medical University. At the time of enrollment, each patient signed an informed consent form in accordance with the specifications stipulated by the Helsinki Committee.

### 2.2. HIFU Treatment

The ultrasound-guided HIFU system has previously been described in detail [16]. All patients in the HIFU group underwent one-session HIFU ablation for breast lesion. The ablated extent consisted of the breast cancer and 1.5–2.0 cm of normal mammary tissue surrounding the lesion. The frequency of HIFU transducer was 1.6 MHz, and acoustic focal peak intensities varied from 5000 to 15,000 W cm^−2^. Total HIFU exposure time ranged from 5 min to 2.5 h (mean: 1.3 h).

### 2.3. Surgery

Modified radical mastectomy is a traditional surgical option for breast cancer patients. It involves the removal of the entire breast tissue and most axillary lymph nodes (ALNs). In the present study, a modified radical mastectomy was carried out in 25 patients in the control group and 1-2 weeks (mean, 10.5 days) after HIFU treatment in 23 patients in the HIFU group. In Europe and North America, breast conserving surgery was a preferred treatment option for patients with early stage breast cancer in the 1990s. However, in China, modified radical mastectomy was still the major surgical option for breast cancer during that period because Chinese patients often had more than 2 cm lesion at the time of diagnosis. Paraffin-embedded ALNs, from the previous trial, were used in this study to investigate the potential HIFU effect on TDLN-mediated immune response in all patients.

### 2.4. Histomorphological Analysis

Formalin-fixed paraffin-embedded tissue from the ALNs was sectioned (4 µm thick) and stained for hemotoxylin and eosin (HE). A comprehensive histologic assessment was made for all lymph nodes with specific reference to the variables of paracortical lymphoid cell hyperplasia (PLCH), sinus histiocytosis (SH) and follicular hyperplasia of the cortical area (FHCA). Briefly, the medullary cords were evaluated for width, and the existence of SH was observed. The cortex was measured for number, size and activity of germinal centers (GCs). The paracortical areas were also examined for size, cellularity, content of large pyroninophilic cells, and mitotic activity. 

Using the standardized criteria previously published by the World Health Organization in 1972 [5] for reporting human lymph node morphological characteristics in relation to immunological reactions, all TDLNs were classified in one of the following immunologic reactivity patterns: (1) cellular immune response (lymphocyte predominance), when there was the development of PLCH and SH; (2) humoral immune response (germinal center predominance), when increased numbers of GC extended throughout the cortex and paracortex; (3) unstimulated status, when the lymph node had no significant changes in architecture and cell distribution; (4) lymphocyte depletion, when the lymph node was fibrotic with decreased numbers of lymphocytes and the absence of GC. Each patient was classified in the pattern that was present in the majority of the nodes. If both cellular and humoral immune patterns occurred simultaneously in an individual node, a mixed immune response was designated for this lymph node. Immunomorphological responses were assessed by two independent observers who were blinded to the clinical data and treatment methods. For 95% of the lymph node slides, the two observers’ assessments concurred. The remaining samples (5%) were re-evaluated together, and consensus conclusions were made after discussions. The number of the ALNs examined for each patient ranged from three to eight with an average of five. Massive metastatic ALNs and poor-quality specimens were excluded from this analysis because of difficulty in making an evaluation. Based on ALNs’ immune reactivity, the patients were divided into one of the following groups: patients with a predominantly humoral immune response, patients with a predominantly cellular immune response, patients exhibiting mixed cellular and humoral immune responses, and patients with no evident immune response that included unstimulated reaction and lymphocyte depletion.

### 2.5. Immunohistochemical Staining

Tissue sections, 4 µm thick, were cut from formalin-fixed, paraffin-embedded ALN tissues. After antigen retrieval, a standard biotin-streptavidin-peroxidase immunohistochemical staining was applied to measure CD3, CD4, CD8, CD20 and CD57 cells in all ALNs. This stain was also carried out to assess the expression of granzyme (GzB), Fas ligand (FasL) and perforin (Pf) on CTLs and NK cells, respectively. The detail of primary antibodies used for immunostaining are listed in Appendix A.

After ALN sections were dewaxed in xylene and dehydrated in a graded alcohol, 3% hydrogen peroxide was used to block endogenous peroxidase activity. The primary antibodies were incubated with the specimens at room temperature, and a biotinylated secondary antibody was subsequently incubated with the specimens. The chromogen was 3,3-diaminobenzidine, It was applied for peroxidase reaction and the slides were counterstained with hematoxylin.

The positively stained cells were considered when homogeneous brown staining was clearly visible. Each slide was reviewed for representative areas of the staining. The number of positively immunostained cells was assessed by two examiners who worked independently. As the lymphocytes were heterogeneosly distributed, the whole slide was first scanned at high powered fields (×400 magnification). Ten areas with the highest number of the stained-positive cells were then selected in each section. The density of the stained-positive cells was counted in a microscopic grid which size is 0.5 × 0.5 mm (0.25 mm^2^), including CD3^+^, CD4^+^, CD8^+^, CD20^+^, and CD57^+^ cells, as well as FasL^+^, GzB^+^ and Pf^+^ CTLs. The counts are presented as the average number of cells per square mm.

### 2.6. Statistical Analysis

Categorical variables differences were tested using the Chi-square test. Continuous variables were expressed as mean ± standard deviation. Using the unpaired Student’s t-test, statistical differences were evaluated between the mean values of the control and HIFU groups. A *p* value less than 0.05 was considered statistically singificant. 

## 3. Results

### 3.1. Clinical and Pathological Characteristics of the Patients with Breast Cancer

After pathological confirmation, 48 breast cancer patients aged from 23 to 70 years were enrolled in the trial. As shown in Appendix A, of 25 patients in the control group, 12 (48%) had tumor-positive lymph nodes and the remaining 13 (52%) had tumor-free ALNs. In the HIFU group, 12 (52.2%) patients had tumor-positive ALNs and the remaining 11 (47.8%) had tumor-negative lymph nodes. Among our study population infiltrating ductal carcinoma was the most frequent tumor type, including 19/25 (79%) in the control group and 21/23 (81%) in the HIFU group. According to TNM staging American Joint Committee on Cancer (AJCC), 6th edition, 22/25 (88%) patients in the control group and 21/23 (91%) patients in the HIFU group were in Stage II. At the time of diagnosis none of them had distance metastasis (Stage IV). 

### 3.2. Alterations in the Immune Response of TDLNs after HIFU Treatment

HIFU could significantly enhance immune reactivity in the ALNs. As shown in Table 1, 64% (16/25) of patients presented an immune response and 36% (9/25) of patients presented no immune response in the control group, whereas in the HIFU group 100% (23/23) of patients showed an immune response. There was a significant difference (*p* < 0.001) between the two groups. In addition, HIFU could also change immune response patterns in the ALNs. In the HIFU group, 78.3% (18/23) of patients presented mixed cellular and humoral immune response, whereas in the control group only 8% (2/25) of patients had the mixed response. A significant difference (*p* < 0.001) was statistically observed between the two groups. However, a cellular immune response pattern occurred more frequently in the control group (36%) than in the HIFU group (13%). A similar result was also observed in humoral immune response between the HIFU group (8.7%) and control group (20%).

### 3.3. HIFU Increases Profile of Immune Cells in TDLNs

Using immunohistochemical staining, T lymphocyte and subsets, B lymphocyte and NK cell populations were examined in all TDLNs. As shown in Figure 1, CD3^+^, CD4^+^ and CD8^+^ T lymphocytes were predominantly present in the paracortical zones and the interfollicular areas of the cortex, although there was a variation in the degree of their overall distributions among individual samples. Compared to the control, CD3^+^ distribution areas were significantly expanded with more nodular hyperplasia in the HIFU group. In addition, CD4^+^ cells appeared among the secondary lymphoid follicles, and CD3^+^ and CD8^+^ cells also appeared in the medullary sinus in the HIFU group. CD20^+^ cells were located in the secondary lymphoid follicles and GCs, as well as the paracortical area. CD57^+^ cells were mainly present in the paracortical area. They appeared mostly in a dispersed distribution in the control groups, whereas more clustered CD57^+^ cells were observed in the HIFU group.

The number of immune cells was evaluated in the paracortical area of both tumor-involved and tumor-free ALNs in the two groups. As shown in Figure 2, CD3^+^, CD4^+^ and CD57^+^ cells were more frequent in the HIFU group while compared to the control group. There were significant differences between the control and HIFU groups for CD3^+^ (*p* < 0.001), CD4^+^ (*p* < 0.001) and CD57^+^ cells (*p* = 0.001). The ratio of CD4^+^ to CD8^+^ cells was 2.41 ± 1.37 in the control group and 3.51 ± 1.49 in the HIFU group. A significant difference (*p* = 0.01) was found between the two groups. However, slightly increased numbers of CD8^+^ cells and decreased numbers of CD20^+^ cells in the paracortex were observed in the HIFU group as compared with the control, but these did not reach statistical differences.

### 3.4. HIFU Activates Cytotoxic T Lymphocytes in TDLNs

FasL^+^, GzB^+^ and Pf^+^ T cells were mainly present in the paracortical area (Figure 3). They also appeared in a dispersed distribution around tumor cells in the metastasis-positive ALNs. Although both activated CTLs and cancer cells expressed FasL, a clear morphological difference was presented between the tumor cells and FasL-positive CTLs in histology. Tumor cells were larger and presented as tumor “nests” or “islands”, while T cells were diffusely scattered, displaying local inflammatory response outside the tumor “nests”. Compared to the control, some FasL^+^, GzB^+^ and Pf^+^ lymphocytes were present in a cluster and they also appeared in the medullary sinus and cord in the HIFU group.

Both tumor-involved and tumor-free ALNs were evaluated to determine the frequency of FasL^+^, GzB^+^ and Pf^+^ T cells in the paracortical arear in both groups. As shown in Figure 4, the number of the FasL^+^, GzB^+^ and Pf^+^ CTLs was significantly higher in the HIFU group. The differences in FasL^+^ (*p* < 0.001), GzB^+^ (*p* < 0.005), and Pf^+^ (*p* < 0.001) CTLs were statistically significant between the control group and HIFU group.

## 4. Discussion

TDLNs are small peripheral lymphoid compartments that lie directly downstream of tumors. They are the most critical site where tumor antigens are typically first presented to the naïve immune system [17]. During the early stage of most cancers, APCs interact with T lymphocytes in the TDLNs to initiate and propagate cellular immune response. Therefore, TDLNs’ immune status influences local tumor microenvironment, and discovering the immune status of the TDLNs may help to understand the host immune status [18]. Thus, local microenvironment in the TDLNs becomes a key determinant in setting the course of the subsequent antitumor immune response [19,20]. To our knowledge, this is the first study using paraffin-embedded ALN samples from breast cancer patients to demonstrate profound alterations in the immune response pattern and immune cell profile of the TDLNs after HIFU treatment. The most significant finding will provide robust evidence for understanding the mechanism of HIFU immunomodulation.

Due to the presence of upstream tumors, TDLNs initiate and develop complex immune reactions in response to antigenicity of tumor cells. They may simultaneously serve as efficient barriers to destroy invading tumor cells, or at least stop their dissemination temporarily. The immune reaction leading to profound morphologic alterations is reflective of a specific immune response pattern in the TDLNs. It can be histologically assessed to demonstrate which immune response is predominant. The morphologic change that indicates a humoral immune response is the presence of FHCA, and the morphologic features that are considered to reflect a cellular immune response are the development of either SH or PLCH, or both [3,4,5,6]. In this study we found that the TDLNs presented more evident immune reactions in the HIFU group than in the control group (100% vs. 64%). Of those in the HIFU group, the majority of patients (78.3%) presented a mixed cellular and humoral immune response, whereas 36% patients showed cellular immune response in the control group (Table 1). There were significant differences in both immune responses and immunological patterns between the HIFU and control groups. Our results indicate that HIFU ablation for primary breast cancer can stimulate potent immune reactions in the TDLNs, which is characterized by the predominant pattern of mixed cellular and humoral immune response.

The paracortex is a T-cell zone in the TDLNs. It contains high endothelium venules that work as a place where APCs encounter specific T lymphocytes for actively presenting tumor antigens. To determine whether the profile of immune cells was different in the TDLNs after HIFU treatment, we subsequently used immunohistochemical staining to analyze T cell (CD3+, CD4+, CD8+), B cell (CD20+) and NK (CD57+) cell populations in the paracortical areas. We found a significant increase of the CD4^+^/CD8^+^ ratio in the HIFU group (*p* = 0.01) while compared with the control group. The frequency of CD3^+^, CD4^+^ and NK cells was also significantly higher in the HIFU group than in the control group (Figure 2). These results indicate that HIFU ablation for primary breast cancer can significantly enhance the cellular immune activity of the TDLNs. It induces not only the increase of T-cell and NK cell populations, but also the regulation of T cell-mediated immunity in the TDLNs.

The most remarkable change observed in this study was that activated CTLs and NK cell populations were significantly increased in the TDLNs following ultrasound treatment (Figure 4). As constant companions, CTLs and NK cells are major cytotoxic effector cells in the immune system. They are crucial mediators of cellular antitumor immunity. The mechanisms that CTLs and NK cells use to attack tumor cells consist of the secretion of death-inducing effector molecules towards the target cell, most importantly GzB, Pf and FasL, as well as the production of effector cytokines such as tumor necrosis factor. They both directly trigger typical CTLs-induced lysis and apoptosis of cancer cells through either FasL or - GzB/Pf- mediated pathways [21,22,23]. Although both cytotoxic cells act upon tumor cells in the same way by inducing apoptosis, the processes involved in their mediated-cell recognition are different. Prior to attacking tumor cells, CTLs require activation by APC processing and presenting specific tumor antigen through MHC class or CD1 molecules [24]. In contrast, NK cells destroy tumor cells without any APC activation, and their activity occurs independently of the presented specific tumor antigen [25]. Our results show increased numbers of activated cytotoxic effector cells with FasL^+^, GzB^+^ and Pf^+^ expressions respectively after HIFU ablation, indicating that HIFU can significantly stimulate lymphocyte activation and subsequently drive the enhanced cellular immune activity in the TDLNs.

TDLNs are the immunologically active sites where APCs interact with T cells in the paracortical area and initiate antigen presentation and lymphocyte activation. After antigen-receptor-mediated activation, CD8 lymphocytes begin to proliferate and differentiate into CTLs that provides immune protection against tumor cells. Therefore, the immune reaction of TDLNs is emerging as a key element in determining the subsequent development of antitumor immune response. Unfortunately, recent studies showed that TDLNs induce immune suppression status in cancer patients with the characteristics of accumulation of immunosuppressive cells such as regulatory T cells (Tregs) and priming of dysfunctional antitumor T cells [26,27]. However, the present study shows that HIFU ablation for primary tumors can present a favorable environment for developing mature CTLs and NK cells and, most importantly, HIFU can relieve tumor-generated immune suppression and counterattacks to NK cells and CTLs in the TDLNs.

The ability of HIFU ablation to trigger antitumor immune response has been investigated for two decades. Our previous study showed that HIFU could induce obvious infiltration of T cell and subsets, B cells and NK cells surrounding ablated breast lesion, with a significant increase of FasL^+^, GzB^+^ and Pf^+^ TILs [15]. Using these old ALN samples from the same patient group, in this study we found similar cellular immune responses in the TDLNs, manifested by the significant increase of CD3^+^, CD4^+^ and NK cell populations that presented FasL, GzB and Pf molecules. These cells could work as activated CTLs and NK cells and move to both primary tumor and metastasis sites via circulation to induce immunogenic cell death. However, further studies will be needed to explore the migration and activation of macrophage and dendritic cells that subsequently promote tumor antigen presentation to T cells in TDLNs.

Our observations in the present study have an important implication. In the past decade, immune checkpoint inhibition immunotherapy has altered the therapeutic landscape of oncology, but its influence is limited by resistance due to a lack of activated T lymphocytes infiltrating into the tumor [28] and TDLNs [29]. A recent study showed that in a subcutaneous xenograft murine melanoma model, histotripsy could enhance tumor-specific CD8^+^ T cell response in TDLNs and induce abscopal intratumoral CD8^+^ T cell response in distant tumors without any treatment [30]. HIFU combined with gemcitabine could increase immunogenicity in a murine metastatic triple-negative breast cancer model [31]. Furthermore, HIFU upregulated multiple innate immune receptors and immune pathways when combined with checkpoint modulator anti-PD-1 and toll-like receptor 9 agonist CpG in mice with breast cancer [32]. Our clinical studies showed that HIFU could significantly increase TILs, activated CTLs and NK cell populations along the ablation margins [13] as well as in the paracortex, leading to the relief of immune suppression of the primary tumor and TDLNs. These results provide the fundamental basis for the development of a novel therapeutic approach in which HIFU, as a local intervention, would be combined with systemic immunotherapy such as programmed cell death 1 (PD-1) and cytotoxic T-lymphocyte antigen-4 (CTLA-4) inhibitors to potentiate antitumor efficacy of checkpoint inhibition immunotherapy.

In conclusion, based on our previous study, we showed for the first time that HIFU ablation for primary breast cancer could stimulate strong immune response and significantly increase T cell, activated CTLs and NK cell populations in the TDLNs. This immunomodulatory impact may trigger potent cell-mediated immune response in the TDLNs. However, further studies are needed to investigate the synergistic effects of HIFU combined with checkpoint inhibition immunotherapy for cancer patients.

## Figures and Tables

**Figure 1 cells-10-03346-f001:**
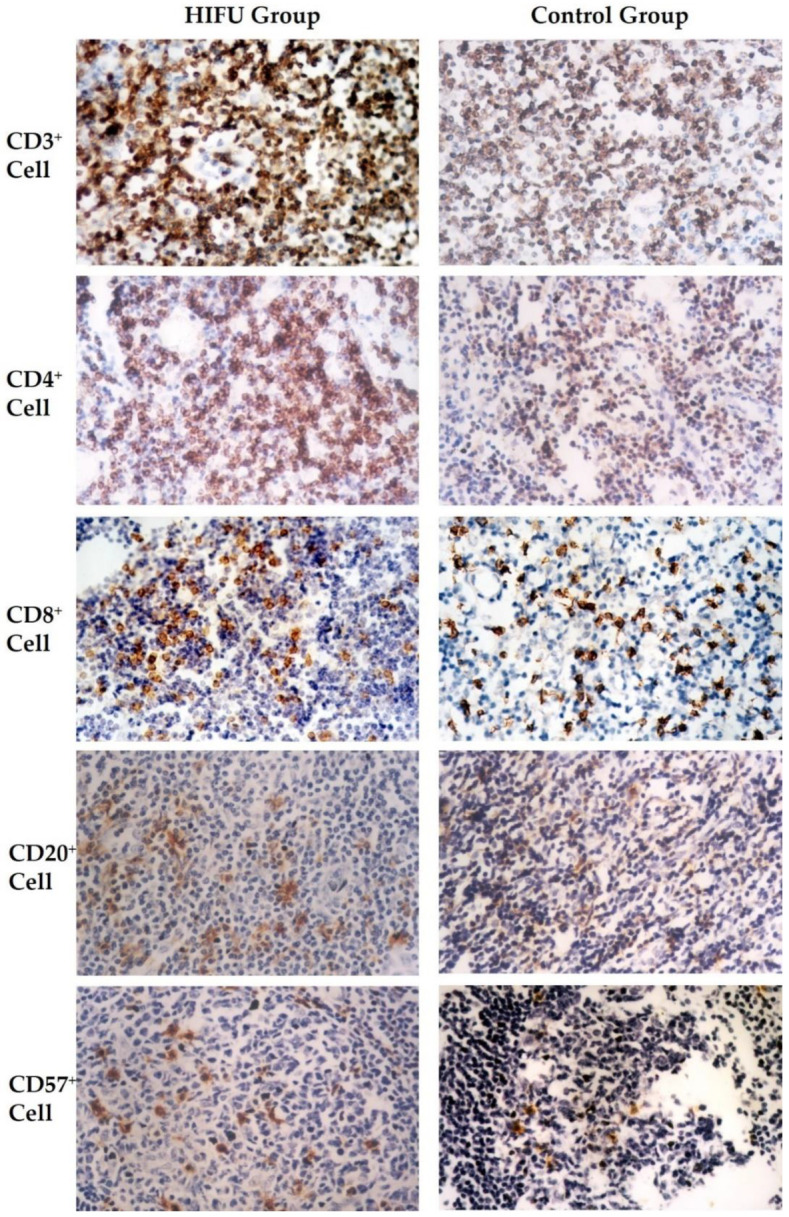
Immunohistochemical staining for formalin-fixed, paraffin-embedded axillary lymph nodes of breast cancer patients. Positive expression of the stained cells is displayed in brown color in the HIFU group (*left*) and control group (*right*), including CD3 (*top row*), CD4 (*second row*), CD8 (*third row*), CD20 (*fourth row*), and CD57 cells (*bottom row*) Streptavidin-peroxidase immunohistochemical staining, ×400.

**Figure 2 cells-10-03346-f002:**
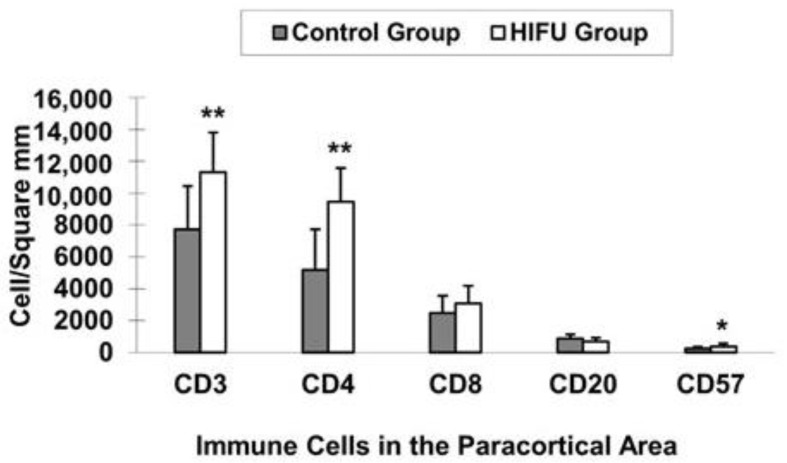
Quantitative analysis of the positively immunostained CD3, CD4, CD8, CD20 and CD57 cells in the paracortical areas of axillary lymph nodes (ALNS) in the HIFU and control groups. Statistically significant difference between the HIFU and control groups: * *p* = 0.001; ** *p* < 0.001.

**Figure 3 cells-10-03346-f003:**
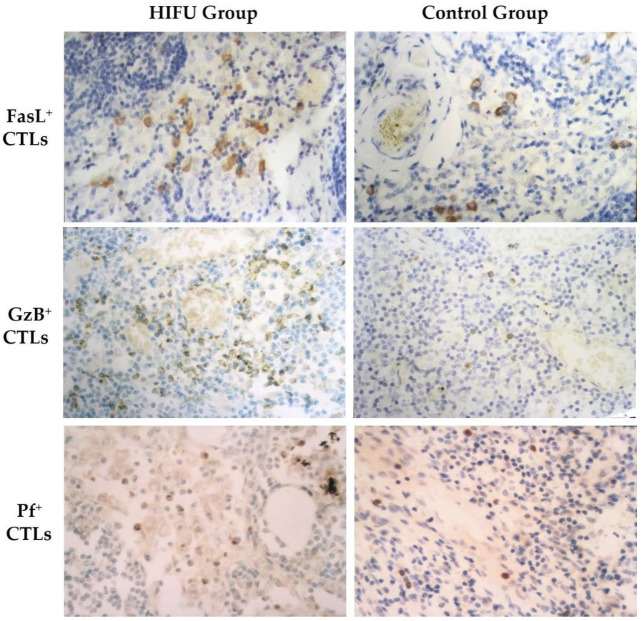
Immunohistochemical staining for formalin-fixed, paraffin-embedded axillary lymph nodes of breast cancer patients. Positive expression of the stained cells is displayed in brown color in the HIFU group (*left*) and control group (*right*), including Fas ligand (FasL^+^) (*top row*), granzyme (GzB^+^) (*second row*), and perforin (Pf^+^) ( *bottom row*) cytotoxic T lymphocytes (CTLs), Streptavidin-peroxidase immunohistochemical staining, ×400.

**Figure 4 cells-10-03346-f004:**
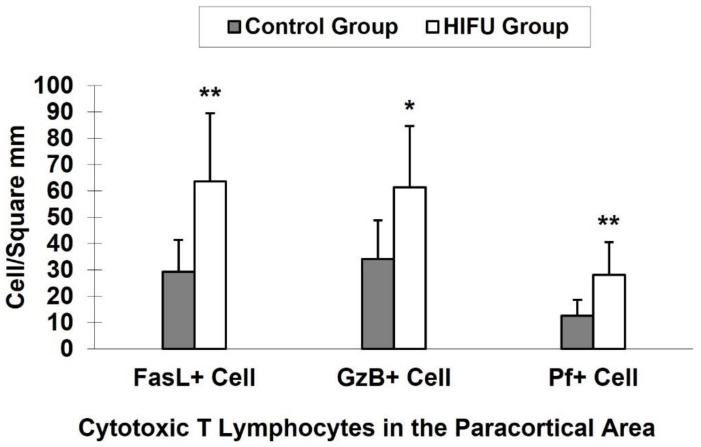
Quantitative analysis of the positively immunostained Fas ligand (FasL^+^), granzyme (GzB^+^), and perforin (Pf^+^) cytotoxic T lymphocytes in the HIFU and control groups. Statistically significant difference between the HIFU and control groups: * *p* < 0.005; ** *p* < 0.001.

**Table 1 cells-10-03346-t001:** Immune reactivity of TDLNs in the control and HIFU groups.

TDLNs Immunoreactivity Pattern	Control Group (*n* = 25)	HIFU Group (*n* = 23)
No immune response ^#^	9 (36%)	0 (0%) *
Immune response	16 (64%)	23 (100%) *
Cellular & humoral immune response	2 (8%)	18 (78.3%) *
Cellular immune response	9 (36%)	3 (13%) *
Humoral immune response	5 (20%)	2 (8.7%)

^#^ No immune response includes unstimulated status and lymphocyte depletion; * Statistical difference is significant between the high-intensity focused ultrasound (HIFU) and control groups.

## Data Availability

The data presented in this study are available on request from the corresponding author. The data of the patients are not publicly available due to privacy rules.

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
