# Peer review of "Alterations in Immune Response Profile of Tumor-Draining Lymph Nodes after High-Intensity Focused Ultrasound Ablation of Breast Cancer Patients"

_cells, 2021, doi:10.3390/cells10123346_

Round 1

Reviewer 1 Report

This is a purely observational study on 20-year old tissues showing increased lymphocytes infiltrate in lymph nodes following an aggressive local treatment- not surprising at all without any clear clinical correlation. Similar data was already shown by the authors in their previous papers.

Author Response

Our response: I fully understand the reviewer’s concern. I would like to explain that there are huge differences between our previously published article (Ref 13. Lu, P. et al. Increased infiltration of activated tumor-infiltrating lymphocytes after high intensity focused ultrasound ablation of human breast cancer. Surgery 2009, 145, 286-293) and current study.  The previous article focused on immune changes in HIFU-treated PRIMARY BREAST CANCER, but the new article’s focus was on immune changes in AXILLARY LYMPH NODES that were not irradiated by HIFU intervention. Although the patients were from the same group in our early clinical trial 20 years ago, the tissue samples used for the two studies are totally different. As tumour draining lymph nodes (TDNDs) play an important role in host antitumour immunity, it is important to investigate the immune response of TDNDs after HIFU treatment. I would say that the findings from the new paper are totally different from the previously published paper, and no similar findings have been published elsewhere before.

This paper is one of a series of HIFU immunomodulation study and it looks extremely difficult for anyone to cover all post-HIFU immune changes in primary tumour, lymph nodes and blood in one paper. In order to establish the whole image of the immune activities after HIFU treatment, it would be ideal if we can publish our results one by one.

Due to the reviewer’s concern, we have requested the Guest Editor Professor Yona Keisari to examine the potential repetition of the two papers. After comparing the submitted manuscript with the previous article carefully, their reply is as follows:

“Dear Prof. Wu,

We are contacting you to update you on your manuscript entitled "Alterations in Immune Response Profile of Tumor-Draining Lymph Nodes after High Intensity Focused Ultrasound Ablation of Breast Cancer Patients " (ID cells-1358333), which you submitted to Cells in the context of the Special Issue "Cancer Immunology: From Molecular Mechanisms to Therapeutic Opportunities".

Because of a misleading review we mistakenly thought that your current submission seemed to be very similar to an article you previously published in the journal Surgery on 2009 (reference no. 13 on the current submission).  After a thorough examination by our Editors and co-Guest Editors, it was realised that these two articles (the Surgery's paper and the current submission) are effectively distinct and different ones. Therefore, this manuscript is still currently under review and you will be properly notified when the reviewers will reach a final decision.

You can see the current status of your manuscript by using your submission credentials as per website instructions at the following link: https://www.mdpi.com/journal/cells/special_issues/cancer_immunology_mol_mech_therap_opportunities.

With best regards,

The Special Issue Guest Editors

Dr. Fabrizio Mattei

Dr. Carlos Alfaro

Prof. Yona Keisari”

Reviewer 2 Report

Overall Comments:

This paper reports the immune response in the tumor-draining lymph nodes after HIFU treatment of patients with breast tumors. The patients received the HIFU ablation and then modified radical mastectomy at 7-14 days post-HIFU. Immunohistology shows increased T cells and activated CTLs and NK cell populations in the TDLNs of 28 breast cancer. There is a concern that this study was done more than 20 years ago (in 1998-1999), and there are certain limitations. However, given the recent interest and literature in HIFU-induced immune response, this study is still relevant. As modified radical mastectomy is no longer a standard therapy, there may not be opportunities to obtain the immunohistology in the TDLNs in human in the future. Therefore, the data reported in this paper can be a valuable contribution to the literature.

My specific comments are as follows.

Introduction

There are a lot of recent publications on HIFU-induced immune response. It would be helpful to cite the recent papers to show the current relevance of this study.

Method

2.2 HIFU treatment - Did you record the temperature increase of the HIFU treatment? How did you determine the end point of the HIFU treatment?

2.3 Surgery – It is stated that “Modified radical mastectomy is the primary surgical option for breast cancer patients.” This is not true in the US. Is this still true in China?

2.5 Immunohistochemical Staining – “Then, 10 areas with the highest number of stained cells were chosen in each section. The number of CD3+, CD4+, CD8+, CD20+, and CD57+ cells, as well as FasL+, GzB+ and Pf+ CTLs was counted in a microscopic grid, 0.5 × 0.5 mm in size (0.25 mm2). The counts are expressed as the mean number of cells per square mm.” How many slides did you get from each lymph node? Were the 10 areas chosen from one slide or multiple slides?

Results

Do you have histology and/or immunohistology of the ablated tumor? Were there remaining tumor cells in the treatment region? How about T cells, NK cells, myeloid cells, and inflammation in and surrounding the treatment region?

In the HIFU group 78.3% (18/23) patients presented mixed cellular and humoral immune response, whereas in the control group only 8% (2/25) patients had the mixed response. How did you define the mixed response exactly?

Please add a scale bar to Figure 1 and Figure 3.

Did you examine the myeloid cells in the TDLNs?

Discussion

There are some recent nice papers on the HIFU immune response in the mouse breast cancer models, including the immunohistology in the TDLNs. It would be helpful to compare your results to these latest animal studies.

Author Response

My specific comments are as follows.

Introduction

There are a lot of recent publications on HIFU-induced immune response. It would be helpful to cite the recent papers to show the current relevance of this study.

Our response: The authors accept the reviewer’s comment. We have added two new references as follows:

Korbelik, M., Hode, T., Lam, S.S.K., Chen, W.R. Novel immune stimulant amplifies direct tumoricidal effect of cancer ablation therapies and their systemic antitumor immune efficacy. Cells 2021, 10, 492.

Qian, L., Shen, Y., Xie, J., Meng, Z. Immunomodulatory effects of ablation therapy on tumors: Potentials for combination with immunotherapy. Biochim. Biophys. Acta. Rev. Cancer 2020, 1874, 188385.

Method

2.2 HIFU treatment - Did you record the temperature increase of the HIFU treatment? How did you determine the end point of the HIFU treatment?

Our response: We would like to explain that as an US-guided HIFU device was used in this study, we didn’t measure and record the temperature rise during HIFU procedure. However, we measured the grey-scale changes of treated cancer in real-time on B-mode ultrasound imaging. If a hypoechoic region occurred in the focus, we ensured that the tumour had been ablated in the focal area. The end point of the HIFU treatment was that the hypoechoic regions covered the entire targeted tumour.

2.3 Surgery – It is stated that “Modified radical mastectomy is the primary surgical option for breast cancer patients.” This is not true in the US. Is this still true in China?

Our response: The authors accept the reviewer’s comment. We would like to explain that modified radical mastectomy is still a major option for breast cancer patients in China. Although breast conserving surgery has been significantly increasing in the past 20 years, more than 50% patients with breast cancer still undergo modified radical mastectomy in China. We have changed the sentence “Modified radical mastectomy is the primary surgical option for breast cancer patients.” into “Modified radical mastectomy is a traditional surgical option for breast cancer patients.” on line 90, page 2.

2.5 Immunohistochemical Staining – “Then, 10 areas with the highest number of stained cells were chosen in each section. The number of CD3+, CD4+, CD8+, CD20+, and CD57+ cells, as well as FasL+, GzB+ and Pf+ CTLs was counted in a microscopic grid, 0.5 × 0.5 mm in size (0.25 mm2). The counts are expressed as the mean number of cells per square mm.” How many slides did you get from each lymph node? Were the 10 areas chosen from one slide or multiple slides?

Our response: The authors accept the reviewer’s comment. We would like to explain that each slide usually had 3-5 axillary lymph nodes, and  they presented similar immune changes in a patient. We selected 10 areas presenting the most obvious immune changes from from one slide.

Results

Do you have histology and/or immunohistology of the ablated tumor? Were there remaining tumor cells in the treatment region? How about T cells, NK cells, myeloid cells, and inflammation in and surrounding the treatment region?

Our response: The authors accept the reviewer’s comment. We would explain that we have published two articles related to this comment before. They are shown in the References Section as follows:

  1. Lu, P.; Zhu, X.Q.; Xu, Z.L.; Zhou, Q.; Zhang, J.; Wu, F. Increased infiltration of activated tumor-infiltrating lymphocytes after high intensity focused ultrasound ablation of human breast cancer. Surgery 2009, 145, 286-293.
  2. Wu, F., Wang, Z. B.; Cao, Y. D.; Chen, W. Z.; Bai. J.; Zou, J. Z.; Zhu, H. A randomised clinical trial of high-intensity focused ultrasound ablation for the treatment of patients with localised breast cancer. Br. J. Cancer 2003, 89, 2227-2233.

In the HIFU group 78.3% (18/23) patients presented mixed cellular and humoral immune response, whereas in the control group only 8% (2/25) patients had the mixed response. How did you define the mixed response exactly?

Our response: We would like to explain that both cellular and humoral immune responses had unique histological characteristics. The cellular response presented sinus histiocytosis and paracortical lymphoid cell hyperplasia, and the humoral had increased numbers of germinal centres extending to the paracortex. If both cellular and humoral immune patterns were observed in a lymph node, it was classified as a mixed immune response.

Please add a scale bar to Figure 1 and Figure 3.

Our response: The authors fully understand that it is important to use a scale bar in measuring gross tissue samples in macroscopic examination. We would like to explain that both Figure 1 and Figure 3 were the pictures taken under light microscope, and it’s essential to show the required magnification of the light microscope rather than a scale bar.  This is a routine way used in the medical study. The magnification range usually extends from ×10 to ×1000 in medical literatures to observe histological slides under light microscope. In this study magification at 400 was used in Figure 1 and 3 (please see the figure legends).

Did you examine the myeloid cells in the TDLNs?

Our response: Unfortunately, the myeloid cells were not examined in our study. We will definitely do it in the future.

Discussion

There are some recent nice papers on the HIFU immune response in the mouse breast cancer models, including the immunohistology in the TDLNs. It would be helpful to compare your results to these latest animal studies.

Our response: The authors accept the reviewer’s comment. We have added 3 sentences on line 347, page 10 as follows:

“A recent study showed that in a subcutaneous xenograft murine melanoma model, histotripsy could enhance tumor-specific CD8+ T cell response in TDLNS and induce abscopal intratumoral CD8+ T cell response in distant tumors without any treatment [31]. HIFU combined with gemcitabine could increases immunogenicity in a murine metastatic triple-negative breast cancer model [32]. Furthermore, HIFU upregulated multiple innate immune receptors and immune pathways when combined with toll-like receptor 9 agonist CpG and checkpoint modulator anti-PD-1 in mice with breast cancer [33]. “

In addition, we have also added 3 references in the References Section on page 13 as follows:

  1. Qu, S., Worlikar, T., Felsted, A.E., Ganguly, A., Beems, M.V., Hubbard, R., Pepple, A.L., Kevelin, A.A., Garavaglia, H., Dib, J., Toma, M., Huang, H., Tsung, A., Xu, Z., Cho, C.S. Non-thermal histotripsy tumor ablation promotes abscopal immune responses that enhance cancer immunotherapy. J. Immunother. Cancer 2020, 8, e000200.
  2. Sheybani, N.D., Witter, A.R., Thim, E.A., Yagita, H., Bullock, T.N.J., Price, R.J. Combination of thermally ablative focused ultrasound with gemcitabine controls breast cancer via adaptive immunity. J. Immunother. Cancer 2020, 8, e001008.
  3. Fite, B.Z., Wang, J., Kare, A.J., Ilovitsh, A., Chavez, M., Ilovitsh, T., Zhang, N., Chen, W., Robinson, E., Zhang, H., Kheirolomoom, A., Silvestrini, M.T., Ingham, E.S., Mahakian, L.M., Tam, S.M., Davis, R.R., Tepper, C.G., Borowsky, A.D., Ferrara, K.W. Immune modulation resulting from MR-guided high intensity focused ultrasound in a model of murine breast cancer. Sci. Rep. 2021, 11, 927.

Reviewer 3 Report

The manuscript "Alterations in Immune Response Profile of Tumor-Draining Lymph Nodes after High Intensity Focused Ultrasound Ablation of Breast Cancer Patients"  by Zhu et al was very interesting. I am excited that scientists are analyzing imune responses of tumor-draining lymph nodes, as I am convinced that this knowledge will help cancer-immunotherapy. 

I have only one minor issue, which is the description of the immune reactivity, and how this was scored. The authors refer to a paper from 1972, but I could not really find a clear description of how their cellular, humoral or mixed response scoring was done. 

Author Response

Our response: The authors fully accept the reviewer’s comment. We would like to explain that this standardized system was published by the World Health Organization in 1972, and it was used for reporting human lymph node morphological characteristics in relation to immunological reactions in cancer patients. Until now, almost all researchers have used it to report the immune response changes in TDLNs when they were involved in this field. Unfortunately, this method did not design a scoring system to assess the changes in TDLNs, but it focused on the development of unique histological characteristics of cellular immune response (such as sinus histiocytosis and paracortical lymphoid cell hyperplasia), and humoral immune response (such as increased numbers of germinal centres extending to the paracortex) in TDLNs. If both cellular and humoral immune patterns were observed in a lymph node, it was classified as a mixed immune response.

Of course, a new proposed system needs to be developed to replace the old one. I believe that it will be available near soon as immunotherapy has been widely used in clinical practice and human TDLNs is becoming a new target organ for research today. 

Round 2

Reviewer 1 Report

This is an interesting study, suggesting activation of the immune response in the lymph nodes following local treatment of the primary tumor with high intensity focused ultrasound. The study was conducted 20 years ago and since that time HIFU has not become a standard of care, so the clinical relevance of the data is limited. However, due to the development of immunotherapies, not available 20 years ago, there may be some renewed interest in local treatments that can activate immune response. I would suggest the authors to add a paragraph to the discussion and compare their observations in the primary tumor to those in the lymph nodes. They should also refer to the technical issue of studying 20-year old tissues. 

Author Response

The ability of HIFU ablation to trigger antitumor immune response has been investigated for two decades. Our previous study showed that HIFU could induce obvious infiltration of CD3+, CD4+, CD8+, B lymphocytes, and NK cells surrounding the ablated breast lesion, with significantly increased number of FasL+, granzyme+ and perforin+ TILs [15]. Using these old ALNs samples from the same patient group, in this study we found similar cellular immune response in the TDLNs, manifested by the significant increase of CD3+, CD4+ and NK cell populations that presented FasL, GzB and Pf molecules. These cells could work as activated CTLs and NK cells and moved to both primary tumor and metastasis sites via circulation to induce immunogenic cell death. However, further studies will be needed to explore the migration and activation of macrophage and dendritic cells that subsequently promote tumor antigen presentation to T cells in TDLNs.